# In situ imaging of bacterial outer membrane projections and associated protein complexes using electron cryo-tomography

Mohammed Kaplan[1], Georges Chreifi[1], Lauren Ann Metskas[1], Janine Liedtke[2], Cecily R Wood[3], Catherine M Oikonomou[1], William J Nicolas[1], Poorna Subramanian[1], Lori A Zacharoff[4], Yuhang Wang[1], Yi-Wei Chang[5], Morgan Beeby[6], Megan J Dobro[7], Yongtao Zhu[8], Mark J McBride[9], Ariane Briegel[2], Carrie L Shaffer[3,10,11], Grant J Jensen[1,12]*

[1]Division of Biology and Biological Engineering, California Institute of Technology, Pasadena, United States; [2]Leiden University, Sylvius Laboratories, Leiden, Netherlands; [3]Department of Veterinary Science, University of Kentucky, Lexington, United States; [4]Department of Physics and Astronomy, University of Southern California, Los Angeles, United States; [5]Department of Biochemistry and Biophysics, Perelman School of Medicine, University of Pennsylvania, Philadelphia, United States; [6]Department of Life Sciences, Imperial College London, London, United Kingdom; [7]Hampshire College, Amherst, United States; [8]Department of Biological Sciences, Minnesota State University, Mankato, United States; [9]Department of Biological Sciences, University of Wisconsin-Milwaukee, Milwaukee, United States; [10]Department of Microbiology, Immunology, and Molecular Genetics, University of Kentucky, Lexington, United States; [11]Department of Pharmaceutical Sciences, University of Kentucky, Lexington, United States; [12]Department of Chemistry and Biochemistry, Brigham Young University, Provo, United States

*For correspondence:
jensen@caltech.edu

**Competing interest:** The authors declare that no competing interests exist.

**Abstract** The ability to produce outer membrane projections in the form of tubular membrane extensions (MEs) and membrane vesicles (MVs) is a widespread phenomenon among diderm bacteria. Despite this, our knowledge of the ultrastructure of these extensions and their associated protein complexes remains limited. Here, we surveyed the ultrastructure and formation of MEs and MVs, and their associated protein complexes, in tens of thousands of electron cryo-tomograms of ~90 bacterial species that we have collected for various projects over the past 15 years (Jensen lab database), in addition to data generated in the Briegel lab. We identified outer MEs and MVs in 13 diderm bacterial species and classified several major ultrastructures: (1) tubes with a uniform diameter (with or without an internal scaffold), (2) tubes with irregular diameter, (3) tubes with a vesicular dilation at their tip, (4) pearling tubes, (5) connected chains of vesicles (with or without neck-like connectors), (6) budding vesicles and nanopods. We also identified several protein complexes associated with these MEs and MVs which were distributed either randomly or exclusively at the tip. These complexes include a secretin-like structure and a novel crown-shaped structure observed primarily in vesicles from lysed cells. In total, this work helps to characterize the diversity of bacterial membrane projections and lays the groundwork for future research in this field.

## Introduction

Membrane extensions and vesicles (henceforth referred to as MEs and MVs) have been described in many types of bacteria. They are best characterized in diderms, where they stem mainly from the outer membrane (OM; we thus refer to OMEs and OMVs) and perform a variety of functions (*Schwechheimer and Kuehn, 2015*; *Jan, 2017*; *D'Souza et al., 2018*; *Toyofuku et al., 2019*). For example, the OMEs of *Shewanella oneidensis* (aka nanowires) are involved in extracellular electron transfer (*Pirbadian et al., 2014*; *Subramanian et al., 2018*). The OM tubes of *Myxococcus xanthus* are involved in the intra-species transfer of periplasmic and OM-associated material between different cells that is essential for the complex social behavior of this species (*Ducret et al., 2013*; *Wei et al., 2014*; *Remis et al., 2014*). The OMVs of *Vibrio cholerae* act as a defense mechanism, helping the bacterium circumvent phage infection (*Reyes-Robles et al., 2018*). A marine Flavobacterium affiliated with the genus *Formosa* (strain Hel3_A1_48) extrudes membrane tubes and vesicles that contain the type IX secretion system and digestive enzymes (*Fischer et al., 2019*). OMVs often function in pathogenesis. The OM blebs and vesicles of *Flavobacterium psychrophilum* have proteolytic activities that help release nutrients from the environment and impede the host immune system (*Møller et al., 2005*). The OMVs of *Francisella novicida* contain virulence factors, suggesting they are involved in pathogenesis (*McCaig et al., 2013*). Similarly, the virulence of *Flavobacterium columnare* is associated with the secretion of OMVs (*Laanto et al., 2014*), and membrane tubes and secreted vesicles have been observed in other, human pathogens like *Helicobacter pylori* and *Vibrio vulnificus* (*Chang et al., 2018*; *Hampton et al., 2017*).

MEs and MVs are also produced by monoderm bacteria and archaea. MVs stemming from the cytoplasmic membrane of Gram-positive bacteria have been reported to encapsulate DNA (see *Brown et al., 2015* and references therein). Membrane nanotubes were recently discovered in the Gram-positive *Bacillus subtilis*, as well as the Gram-negative *Escherichia coli*. These nanotubes were found to connect two different bacterial cells and are involved in the transfer of cytoplasmic material between bacterial cells of the same and different species, and even to eukaryotic cells (*Bhattacharya et al., 2019*; *Dubey and Ben-Yehuda, 2011*; *Baidya et al., 2018*; *Pande et al., 2015*; *Benomar et al., 2015*; *Baidya et al., 2020*; *Pal et al., 2019*). In addition, a recent study suggested that nanotubes assist the growth of *Pseudomonas aeruginosa* on periodic nano-pillar surfaces (*Cao et al., 2020*).

The structures of MEs and MVs are as varied as their functions. While *S. oneidensis* nanowires are chains of interconnected OMVs with variable diameter and decorated with cytochromes (*Subramanian et al., 2018*), OM tubes of *H. pylori* have a fixed diameter of ~40 nm and are characterized by an inner scaffold and lateral ports (*Chang et al., 2018*). *V. vulnificus* produces tubes from which vesicles ultimately pinch off by biopearling, forming a regular concentric pattern surrounding the cell (*Hampton et al., 2017*). Cells with an external surface layer (S-layer) can produce structures known as 'nanopods', which consist of MVs inside a sheath of S-layer. These have been reported in the soil-residing bacterium *Delftia* sp. Cs1–4 (*Shetty et al., 2011*) and archaea of the order Thermococcales (*Marguet et al., 2013*). Finally, some diderms produce DNA-containing MVs consisting of both IM and OM (see *Toyofuku et al., 2019* and references therein).

Different models have been proposed for how MEs and MVs form. In diderms, membrane blebbing may occur due to changes in the periplasmic turgor pressure, lipopolysaccharide repulsion, or alterations in the contacts between the OM and the peptidoglycan cell wall (*Toyofuku et al., 2019*). Chains of interconnected vesicles are often observed, either as a result of direct vesicular budding from the OM or due to biopearling of membrane tubes (*Subramanian et al., 2018*; *Fischer et al., 2019*). Formation of tubes is thought to be a stabilizing factor as it results in smaller vesicles, with tubes pearling into distal chains of vesicles that eventually disconnect (*Bar-Ziv and Moses, 1994*). Other extensions may be formed by dedicated machinery. Interestingly, nanotubes involved in cytoplasmic exchange have been reported to be dependent on a conserved set of proteins involved in assembly of the flagellar motor known as the type III secretion system core complex (CORE): FliP/O/Q/R and FlhA/B (*Bhattacharya et al., 2019*; *Pal et al., 2019*). Recently, it was also shown that the formation of bacterial nanotubes significantly increases under stress conditions or in dying cells, caused by biophysical forces resulting from the action of the cell wall hydrolases LytE and LytF (*Pospíšil et al., 2020*).

Structural studies of MEs and MVs have relied mainly on scanning electron microscopy (SEM), conventional transmission electron microscopy (TEM), and light (fluorescence) microscopy. While these methods have significantly advanced our understanding, they are limited in terms of the

information they can provide. For instance, in SEM and conventional TEM, sample preparation such as fixation, dehydration, and staining disrupt membrane ultrastructure. While light microscopy can reveal important information about the dynamics and timescales on which MEs and MVs form (e.g. *Bos et al., 2021*), no ultrastructural details can be resolved; MEs and MVs of different morphology appear identical. Currently, only electron cryo-tomography (cryo-ET) allows visualization of structures in a near-native state inside intact (frozen-hydrated) cells with macromolecular (~5 nm) resolution. However, this capability is limited to thin samples (few hundred nanometers thick, like individual bacterial cells of many species) while thicker samples like the central part of eukaryotic cells, thick bacterial cells, or clusters of bacterial cells are not amenable for direct cryo-ET imaging. Such thick samples can be rendered suitable for cryo-ET experiments by thinning them first using different methods including focused ion beam milling and cryosectioning (*Kaplan et al., 2021a*). Cryo-ET has already been invaluable in revealing the structures of several MEs, including *S. oneidensis* nanowires (*Subramanian et al., 2018*), *H. pylori* tubes (*Chang et al., 2018*), *Delftia acidovorans* nanopods (*Shetty et al., 2011*), *V. vulnificus* OMV chains (*Hampton et al., 2017*), and more recently cell-cell bridges in the archaeon *Haloferax volcanii* (*Sivabalasarma et al., 2020*).

To understand what MEs exist in bacterial cells and how they might form, we undertook a survey of ~90 bacterial species, drawing on a database of tens of thousands of electron cryo-tomograms of intact cells collected by our group for various projects over the past 15 years (*Ding et al., 2015*; *Ortega et al., 2019*), in addition to data generated in the Briegel lab. Our survey revealed OM projections in 13 diderm bacterial species. These projections took various forms: (1) tubes with a uniform diameter and with an internal scaffold, (2) tubes with a uniform diameter and without a clear internal scaffold, (3) tubes with a vesicular dilation at their tip (teardrop-like extensions), (4) tubes with irregular diameter or pearling tubes, (5) interconnected chains of vesicles with uniform neck-like connectors, (6) budding or detached OMVs, and (7) nanopods. We also identified protein complexes associated with MEs and MVs in these species. These complexes were either seemingly randomly distributed on the MEs and MVs or exhibited a preferred localization at their tip.

## Results

We examined tens of thousands of electron cryo-tomograms of ~90 bacterial species collected in the Jensen lab for various projects over the past 15 years together with tomograms collected in the Briegel lab. Most cells were intact, but some had naturally lysed. Note that we make this classification based on the cells' appearance in tomograms; intact cells have an unbroken cell envelope, uniform periplasmic width, and consistently dense cytoplasm. In addition to cryo-tomograms of cells, this dataset also included naturally shed vesicles purified from *S. oneidensis*. In all, we identified OMEs and OMVs in 13 bacterial species (summarized in *Table 1*, *Table 2*).

### I – The diverse forms of bacterial membrane structures

Based on their features, we classified membrane projections into the following categories: (1) tubular extensions with a uniform diameter and with an internal scaffold (*Figure 1a and b* and *Figure 1— figure supplements 1 and 2*); (2) tubular extensions with a uniform diameter and without a clear internal scaffold (*Figure 1c–g* and *Figure 1—figure supplement 3*); (3) tubular extensions with a vesicular dilation at the tip (a teardrop-like structure) and irregular dark densities inside (*Figure 1h*); (4) tubular extensions with irregular diameter or pearling tubes (*Figure 2a–g*); (5) interconnected chains of vesicles with uniform neck-like connectors (*Figure 2h & i*); (6) budding or detached vesicles: budding vesicles were still attached to the membrane, while detached vesicles were observed near a cell and could have budded directly or from a tube that pearled (*Figure 3a–d* and *Figure 3—figure supplement 1*); (7) nanopods: tubes of S-layer containing OMVs (*Figure 3e–i*). See *Table 1* for a summary of these observations.

Scaffolded membrane tubes were observed only in *H. pylori* and had a uniform diameter of 40 nm. The *H. pylori* strain imaged (*fliP*\*) contains a naturally occurring point mutation that disrupts the function of FliP, the platform upon which other CORE proteins assemble (*Fukumura et al., 2017*; *Fabiani et al., 2017*; *Minamino et al., 2019*). In addition, the dataset contained other mutants in this *fliP*\* background including additional CORE proteins (Δ*fliO* and Δ*fliQ*), flagellar basal body proteins (Δ*fliM* and Δ*fliG*), and the tyrosine kinase required for expression of the class II flagellar genes (Δ*flgS*)

**Table 1.** A summary of the species included in this study and the major membrane structures identified in each species. Note that the approximation symbol before the number of cells indicates that in many tomograms we only see a part of the cell(s).

| Species | Class | No. of cells | Tubes | | | Pearling | Vesicle chains | | Budding/ vesicles | Nanopods |
|---|---|---|---|---|---|---|---|---|---|---|
| | | | Uniform diameter – scaffold | Uniform diameter – no scaffold | Variable diameter | | Connectors | No connectors | | |
| *Shewanella oneidensis* | Gammaproteobacteria | ~700 | | | | | | See *Subramanian et al., 2018* | > 100 | |
| *Pseudoalteromonas luteoviolacea* | Gammaproteobacteria | ~67 | | ~100 | | ~10 | | | | |
| *Hylemonella gracilis* | Betaproteobacteria | ~105 | | | 3 | 4 | | | 15 | |
| *Delftia acidovorans* | Betaproteobacteria | n.a. | | | | | | | | See *Shetty et al., 2011* |
| *Magnetospirillum magneticum* | Alphaproteobacteria | ~56 | | | | | | | 49 | |
| *Caulobacter crescentus* | Alphaproteobacteria | ~464 | | | | | | | | 53 |
| *Helicobacter hepaticus* | Epsilonproteobacteria | ~28 | | | | 2 | | | | |
| *Helicobacter pylori* | Epsilonproteobacteria | ~883 | >100 | | | | | | >100 | |
| *Myxococcus xanthus* | Deltaproteobacteria | ~2000 | | >100 | | >100 | | | >100 | |
| *Borrelia burgdorferi* | Spirochaetes | ~61 | | 9 | | | 19 | | 16 | |
| *Flavobacterium johnsoniae* | Flavobacteria | ~203 | | ~45 | | ~15 | | | >100 | |
| *Flavobacterium anhuiense* | Flavobacteria | ~49 | | | 5 | 7 | | 4 | >100 (including the teardrop-like extensions) | |
| *Chitinophaga pinensis* | Chitinophagia | ~61 | | | 11 | 12 | | 3 | 81 | |

**Table 2.** The different bacterial strains used in this study.

| Species | Strain | Relevant references |
|---|---|---|
| *Shewanella oneidensis* | MR-1 211,586 | *Subramanian et al., 2018*; *Kaplan et al., 2019a*; *Kaplan et al., 2019b* |
| *Pseudoaltermonas luteoviolacea* | 43,657 | *Shikuma et al., 2014* |
| *Hylemonella gracilis* | ATCC 19624 887,062 | *Kaplan et al., 2020*; *Chen et al., 2011*; *Kaplan et al., 2021b* |
| *Delftia acidovorans* | Cs1-4 80,866 | *Shetty et al., 2011* |
| *Magnetospirillum magneticum* | AMB-1 342,108 | *Cornejo et al., 2016* |
| *Caulobacter crescentus* | NA1000 | *Kaplan et al., 2021c* |
| *Helicobacter hepaticus* | ATCC 51449 235,279 | *Chen et al., 2011* |
| *Helicobacter pylori* | 26,695 | *Chang et al., 2018* |
| *Myxococcus xanthus* | DK1622 | *Chang et al., 2016* |
| *Borrelia burgdorferi* | B31 224,326 | *Briegel et al., 2009*; *Chen et al., 2011* |
| *Flavobacterium johnsoniae* | CJ2618 | This study |
| *Flavobacterium anhuiense* | 98 | *Carrión et al., 2019* |
| *Chitinophaga pinensis* | 94 | *Carrión et al., 2019* |

(*Lertsethtakarn et al., 2011*; *Figures 1a–b and 4*, *Figure 1—figure supplement 1* and *Table 3*). This suggests that the *H. pylori* membrane tubes are unrelated to the CORE-dependent nanotubes that mediate cytoplasmic exchange in *B. subtilis* and other species (*Bhattacharya et al., 2019*; *Pal et al., 2019*).

Previously, *H. pylori* tubes were described as forming in the presence of eukaryotic host cells (*Chang et al., 2018*). Here, however, we observed tubes in *H. pylori* grown on agar plates in the absence of eukaryotic cells, suggesting that they also form in the absence of host cells. We observed some differences, though, from the tubes formed in the presence of host cells: the tube ends were closed, no clear lateral ports were seen, and the tubes were usually straight. While some of these tubes extended more than 0.5 µm, we never observed pearling. However, in some tubes, the internal scaffold did not extend all the way to the tip, and its absence caused the tube to dilate (from 40 nm in the presence of the scaffold to 66 nm in its absence, see *Figure 4f* and examples in *Figure 1—figure supplement 1b and d*). In some cases we also observed tubes stemming from vesicles resulting from cell lysis (*Figure 4f*, *Figure 1—figure supplement 1b and d*, *Figure 1—figure supplement 2*), and dark densities could be seen at the base of many of these tubes associated with vesicles (*Figure 1—figure supplement 1d*).

In *Flavobacterium anhuiense* and *Chitinophaga pinensis*, which are both endophytic species extracted from sugar beet roots, in addition to tubes with irregular diameter and OMVs (*Figure 2g*), tubular extensions with a uniform diameter, and a vesicular dilation (teardrop-like structure) were observed stemming from the sides of the cell in *F. anhuiense* (*Figure 1h*). Interestingly, irregular dark densities were observed inside these teardrop-like extensions (*Figure 1h*). Chains of vesicles connected by neck-like bridges were similarly observed in a single species: *Borrelia burgdorferi*. The bridges were consistently ~14 nm in length and ~8 nm in width. Where chains were seen attached to the OM, a neck-like connection was present at the budding site (*Figure 2h*). Vesicles in each chain were of a uniform size, usually 35–40 nm wide (e.g. *Figure 2i*), but occasionally larger (e.g. *Figure 2h*).

When both tubes and vesicles were observed in the same species, the tubes generally had a more uniform diameter than the vesicles, which were of variable sizes and often had larger diameters than the tubes (*Figure 3—figure supplement 1* and *Figure 3—figure supplement 2*). In addition, when a tube pearled into vesicles, there was no clear correlation between the length of the tube and the initiation point of pearling, with some tubes extending for many micrometers without pearling while other, shorter tubes were in the process of forming vesicles (*Video 1*, *Video 2*, *Video 3*, and *Figure 2*). As usually only one (or part of a) cell is present in the cryo-tomogram, we cannot exclude that differences in the extracellular environments, like the presence of a cluster of cells in the vicinity of the individual cells with pearling tubes, might play a role in this observation. Pearling tubes differ from tubes with irregular diameter by the presence of a deep constriction in some part of the tube,

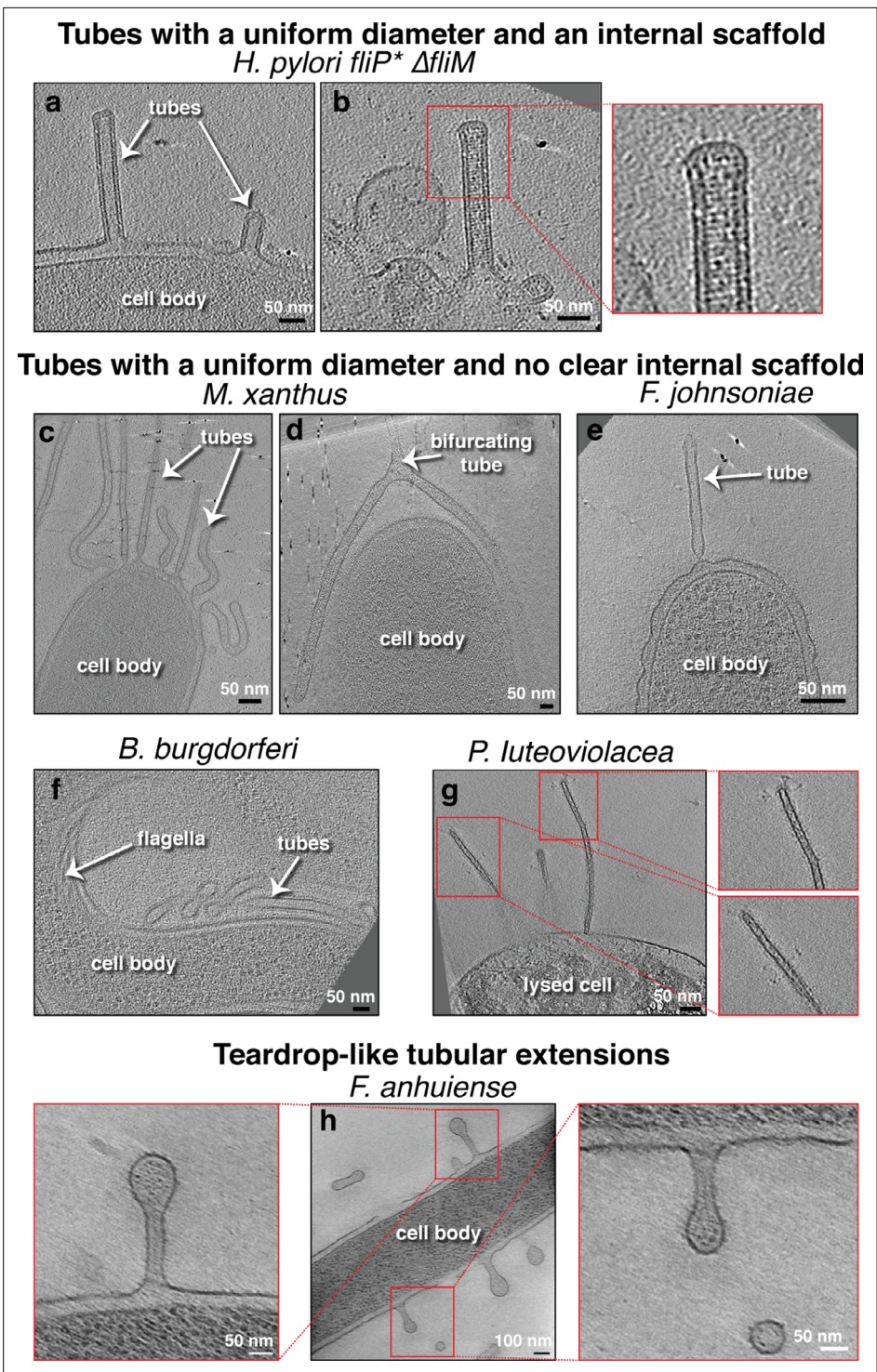

**Figure 1.** Membrane tubes with a uniform diameter, either with or without an internal scaffold. Slices through electron cryo-tomograms of the indicated bacterial species highlighting the presence of outer membrane extensions (OMEs) with uniform diameters and either with (**a–b**) or without (**c–g**) an internal scaffold, and teardrop-like extensions (**h**). In this and all subsequent figures, red boxes indicate enlarged views of the same slice. Scale bars are 50 nm, except in main panel (**h**) 100 nm.

The online version of this article includes the following figure supplement(s) for figure 1:

**Figure supplement 1.** Examples of membrane tubes stemming from intact, lysed or vesicles of *Helicobacter pylori* mutants.

*Figure 1 continued on next page*

*Figure 1 continued*

**Figure supplement 2.** Slices through electron cryo-tomograms of lysed *Helicobacter pylori fliP\* ΔfliM cells* illustrating the presence of outer membrane (OM) tubes in vesicles resulting from cell lysis (black arrows).

**Figure supplement 3.** A slice through an electron cryo-tomogram of a lysed *Pseudoalteromonas luteoviolacea* cell illustrating a bifurcated 20 nm wide membrane tube.

while chains of vesicles are entirely made up of semi-circular vesicles connected by thin constrictions suggesting different mechanisms are responsible for the formation of these different extensions. While most pearling was seen at the tips of tubes, pearling occasionally occurred simultaneously at both proximal and distal ends of the same tube (*Video 3*). With one exception, pearling was seen in all species with tubes of uniform diameter and no internal scaffold. The exception was lysed *Pseudoalteromonas luteoviolacea*, which had narrow tubes only 20 nm in diameter (*Figure 1g*). Some lysed *P. luteoviolacea* contained wider, pearling tubes (*Figure 2c*). Interestingly, the tubes of various *M. xanthus* strains (see Materials and methods) and *P. luteoviolacea* could bifurcate into branches, each of which had a uniform diameter similar to that of the main branch (*Video 4* and *Figure 1d* and *Figure 1—figure supplement 3*).

In *Caulobacter crescentus* tomograms, we identified structures very similar to the 'nanopod' extensions previously reported in *D. acidovorans* (*Shetty et al., 2011*). These structures consist of a tube made of the S-layer encasing equally spaced OMVs (*Figure 3e–h* and *Video 5*). The diameter of the S-layer tubes was ~45 nm and vesicles exhibited diameters ranging from ~13 to 25 nm. The nanopods were seen either detached from the cell (*Figure 3e–g*) or budding from the pole of *C. crescentus* (*Figure 3h*).

## II – Protein complexes associated with membrane structures

Next, we examined protein complexes associated with OMEs and OMVs that we could identify in our cryo-tomograms. These complexes fell into three categories: (1) seemingly randomly located complexes found on OMEs, OMVs, and cells; (2) seemingly randomly located complexes observed only on OMEs and OMVs; and (3) complexes exclusively located at the tip of OMEs/OMVs.

In the first category, we observed what appeared to be the OM-associated portion of the empty basal body of the type IVa pilus (T4aP) machinery in OMEs of *M. xanthus*. These complexes, which were also found in the OM of intact cells, did not exhibit a preferred localization or regular arrangement within the tube at least within the fields of view provided by our cryo-tomograms (*Figure 5*).

The second category of protein complexes, observed only on MEs and not on cells, contained two structures. The first was a trapezoidal structure observed on purified OMVs of *S. oneidensis*. The structure was ~11 nm wide at its base at the membrane and was seen sometimes on the outside (*Figure 5c*) and sometimes the inside of vesicles (*Figure 5d*). The second structure was a large crown-like complex. We first observed these complexes on the outer surface of MVs associated with lysed *M. xanthus* cells (*Figure 6a*). Occasionally, they were also present on what appeared to be the inner leaflet of the inner membrane of lysed cells (*Figure 6b*). The exact topology is difficult to determine, however, since the arrangement of IM and OM can be confounded by cell lysis. The structure of this complex was consistent enough to produce a subtomogram average from nine examples, improving the signal-to-noise ratio and revealing greater detail (*Figure 6c*). These crown-like complexes are ~40 nm tall with a concave top and a base ~35 nm wide at the membrane (*Figure 6c*). No such complexes were seen on OMEs and OMVs associated with intact *M. xanthus* cells. We identified a morphologically similar crown-like complex on the outside of some tubes and vesicles purified from *S. oneidensis* (*Figure 6d–f*). However, this complex was smaller, ~15 nm tall and ~20 nm wide at its base. As these MEs/MVs from *S. oneidensis* were purified, we cannot know whether they stemmed from lysed or intact cells. Interestingly, we found a similar large crown-like structure associated with lysed cells of two other species in which we did not identify MEs, namely *Pseudomonas flexibilis* and *P. aeruginosa* (*Figure 6g–j* and *Figure 6—figure supplement 1*).

In the third category, we observed a secretin-like complex in many tubes and vesicles of *F. johnsoniae*. Secretins are proteins that form a pore in the OM and are associated with many secretion systems like type IV pili and type II secretion systems (T2SS) (*Chang et al., 2016*; *Ghosal et al., 2019*; *Gold et al., 2015*). In tubes attached to the cell, the complex was always located at the distal tip (*Figure 7*, *Figure 7—figure supplement 1*, and *Video 6*). From 35 membrane tubes seen attached

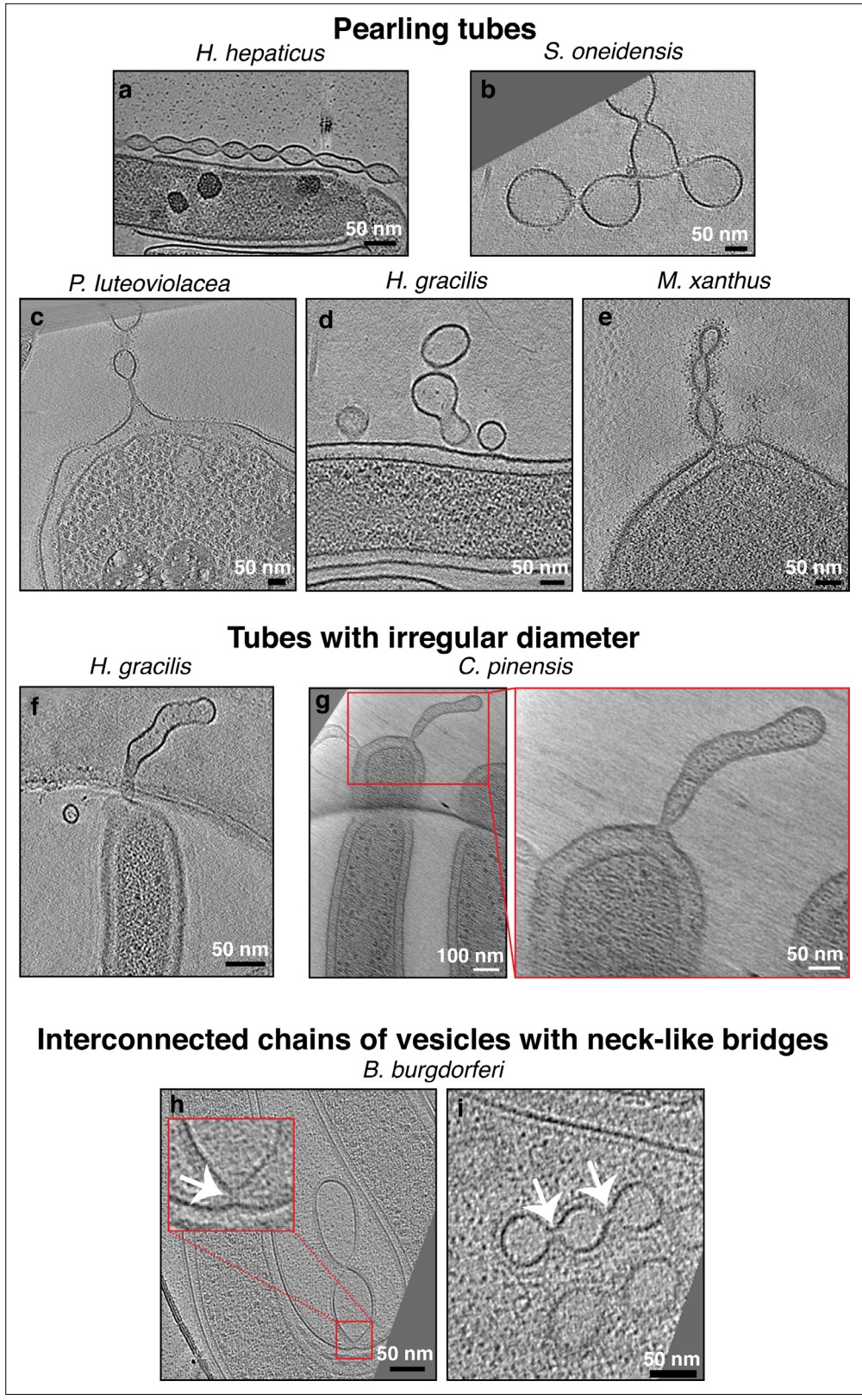

**Figure 2.** Pearling tubes, tubes with irregular diameter, and vesicle chains with neck-like connections. Slices through electron cryo-tomograms of the indicated bacterial species highlighting the presence of pearling tubes (**a–e**), tubes with irregular diameter (**f–g**), or outer membrane vesicle (OMV) chains connected by neck-like bridges (**h–i**). White arrows in the enlargement in (**h**) and in panel (**i**) point to the 14 nm connectors in *Borrelia burgdorferi*. Scale bars are 50 nm, except in main panel (**g**) 100 nm.

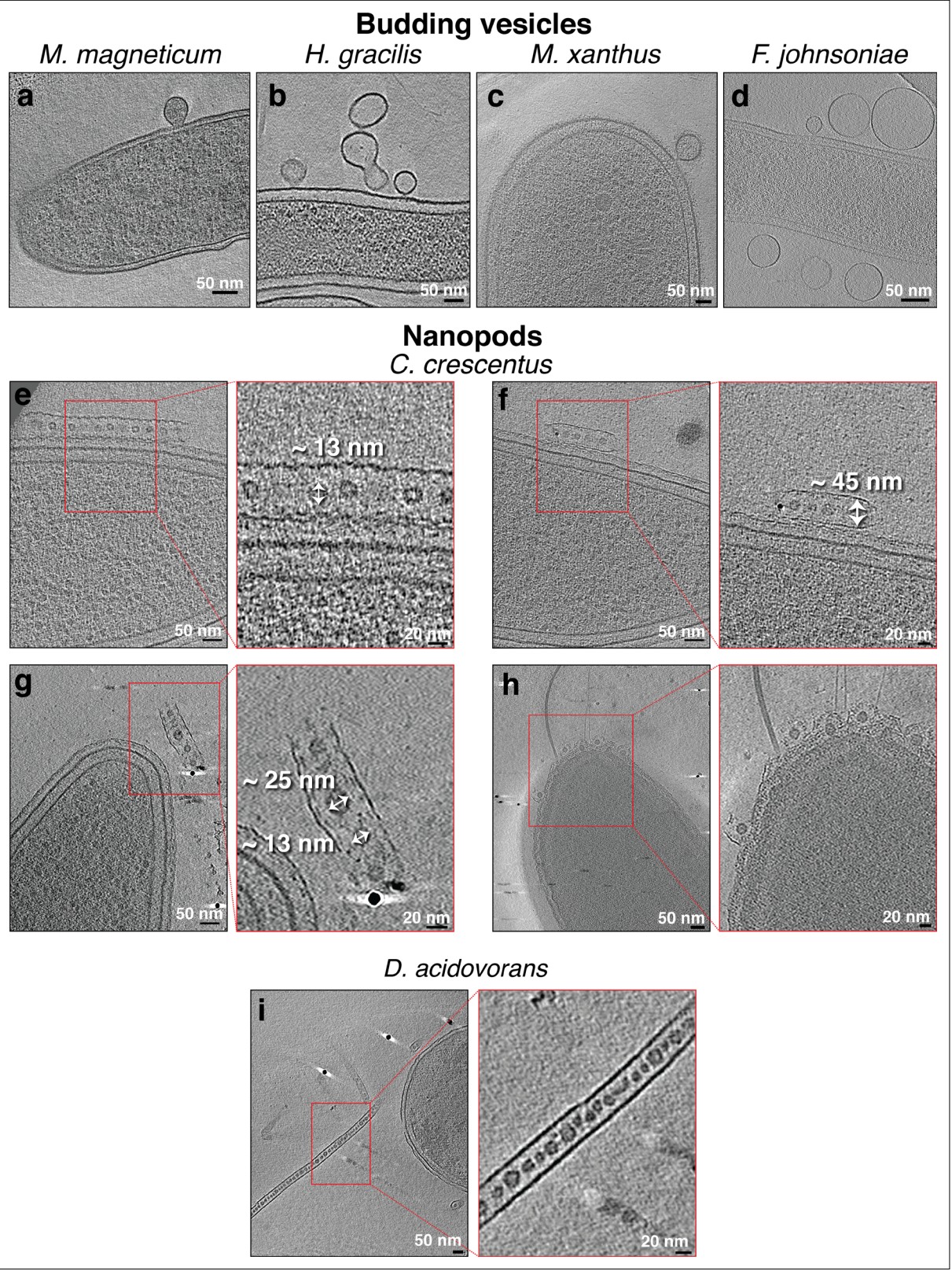

**Figure 3.** Budding outer membrane vesicles (OMVs) and nanopods. Slices through electron cryo-tomograms of the indicated bacterial species highlighting the presence of budding vesicles (a–d) or nanopods (e– i). Scale bars are 50 nm in main panels and 20 nm in enlargements.

The online version of this article includes the following figure supplement(s) for figure 3:

*Figure 3 continued on next page*

*Figure 3 continued*

**Figure supplement 1.** Outer membrane extensions and vesicles in *S. oneidensis* and *M. xanthus*.

**Figure supplement 2.** Violin plots of the sizes of outer membrane (OM) vesicles (OMVs) and OM tubes in *Myxococcus xanthus* (100 randomly picked examples of each) and *Flavobacterium johnsoniae* (45 randomly picked examples of each).

to cells, we identified a secretin-like complex at the tip of 25 of them (~70%). In OMEs disconnected from the cell, the secretin-like complex was always located at one end (*Figure 7b & e*). In total, we identified 88 secretin-like particles in 198 tomograms, none of which were located in the middle of a tube. As the MEs are less crowded than cellular periplasm and usually thinner than intact cells, we could clearly distinguish an extracellular density and three periplasmic densities in side views (red and purple arrows, respectively, in *Figure 7a*). Top views showed a plug in the center of the upper part of the complex (yellow arrows in *Figure 7g & h*). Subtomogram averaging revealed details of the complex, including the plug and a distinct lower periplasmic ring (*Figure 7i & j* & *Figure 7—figure supplement 2*). While the upper two periplasmic rings were clearly distinguishable in many of the individual particles (e.g. *Figure 7a*), they did not resolve as individual densities in the subtomogram average (*Figure 7i*). The extracellular density was not resolved at all in the average, suggesting flexibility in this part.

Previous studies showed that a species which belongs to the same phylum as *F. johnsoniae*, namely *Cytophaga hutchinsonii*, uses a putative T2SS to degrade cellulose (*Wang et al., 2017*). Since *F. johnsoniae* also degrades polysaccharides and other polymers, we BLASTed the sequence of the well-characterized *V. cholerae* T2SS secretin protein, GspD (UniProt ID P45779), against the genome of *F. johnsoniae* and found a hit, GspD-like T2SS secretin protein (A5FMB4), with an e-value of $1e^{-9}$. This result and the general morphological similarity of this secretin to the published structure of the T2SS (*Ghosal et al., 2019*) suggested that the complex we observed might be the secretin of a T2SS. We therefore compared our subtomogram average with the only available in situ structure of a T2SS, a recent subtomogram average of the *Legionella pneumophila* T2SS (*Ghosal et al., 2019*; *Figure 7i–l*). The two structures were generally similar in length and both had a plug in the upper part of the complex. However, we also observed differences between the two structures. In *L. pneumophila*, the widest part of the secretin (15 nm) is located near the plug close to the OM, and the lower end of the complex is narrower (12 nm). In *F. johnsoniae*, this topology is reversed, with the narrowest part near the plug and OM (*Figure 7i–l*). Additionally, the lowest domain of the *L. pneumophila* secretin did not resolve into a distinct ring as we saw in *F. johnsoniae* and no extracellular density was observed in *L. pneumophila*, either in the subtomogram average or single particles (*Ghosal et al., 2019*).

## Discussion

Our results highlight the diversity of MEs' and MVs' structures that bacteria can form even within a single species (*Figure 8*). For example, we saw two types of membrane tubes in lysed *P. luteoviolacea* cells: one narrower with a uniform diameter of 20 nm which did not pearl into vesicles, and one wider with a variable diameter that did pearl into vesicles (*Figures 1g and 2c*), a distinction which suggests that these extensions play different roles. Similarly, interspecies differences likely reflect different functions. For instance, the tubes of *M. xanthus* were on average longer, more abundant, and more branched than the MEs of other species (*Videos 1 and 4*), which is likely related to their role in communication between cells of this highly social species. However, one interesting observation in all the species we investigated here is that there was no clear distinctive molecular machine at the base of the membrane projections, raising the question of what drives their formation. This observation is consistent with a recent study which showed that liquid-like assemblies of proteins in membranes can lead to the formation of tubular extensions without the need for solid scaffolds (*Yuan et al., 2021*). In addition, differences in the lipid compositions among the various species investigated here might also play a role in the formation of these different forms of projections.

The scaffolded uniform tubes of *H. pylori* that we observed were formed in samples not incubated with eukaryotic cells, indicating that they can also form in their absence. However, the tubes we found had closed ends and no clear lateral ports, while some of the previously reported tubes (formed in the presence of eukaryotic host cells) had open ends and prominent ports (*Chang et al., 2018*). It is possible that such features are formed only when *H. pylori* are in the vicinity of host

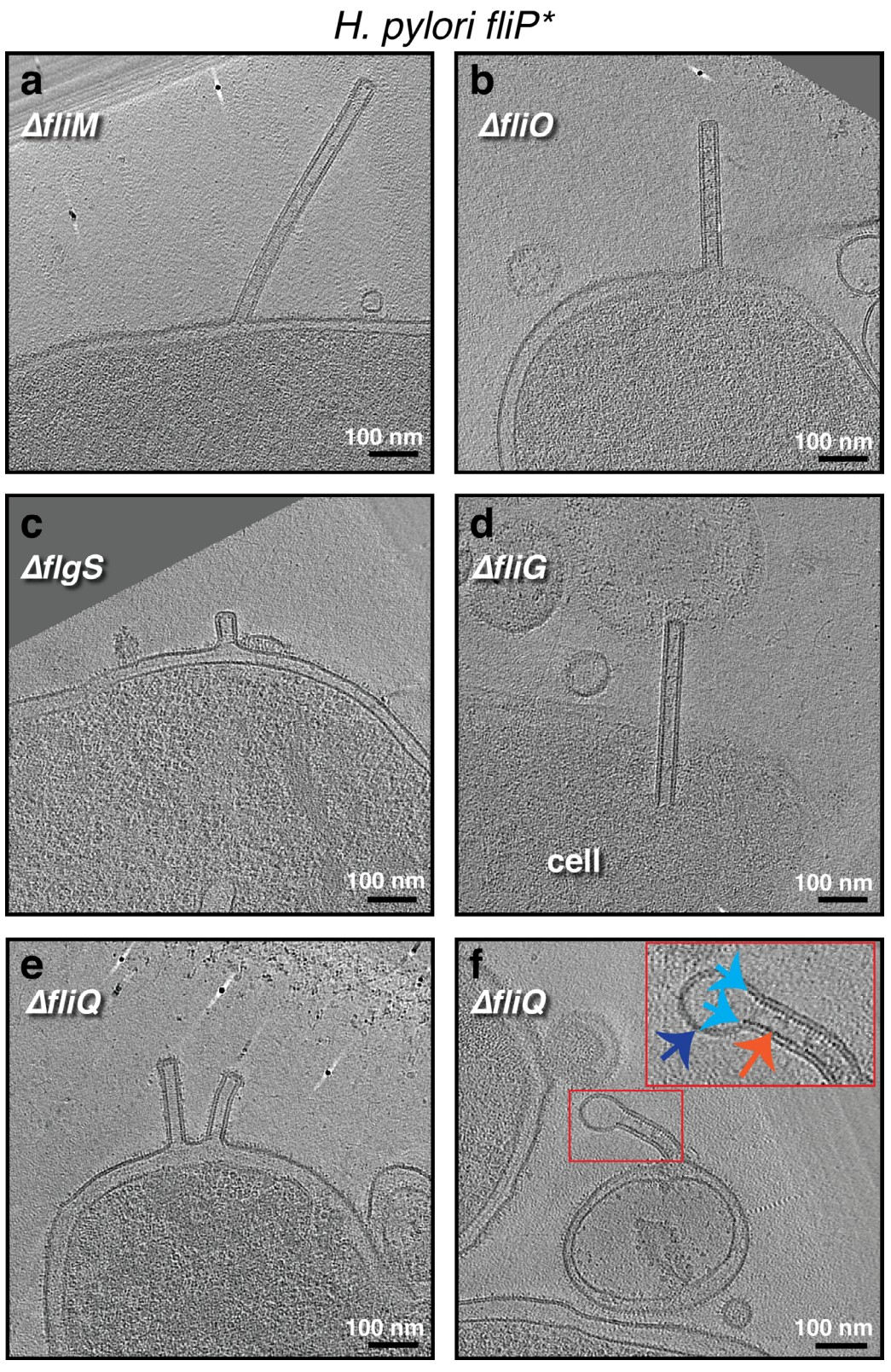

**Figure 4.** The formation of outer membrane (OM) tubes persists in various *Helicobacter pylori* mutants, including CORE mutants. Slices through electron cryo-tomograms of the indicated *H. pylori* mutants (all in the *fliP\** background) showing the presence of membrane tubes. The enlargement in (**f**) highlights a dilation at the end of the tube (dark blue arrow) due to the absence of the scaffold (orange arrow). Light blue arrows indicate the end points of the scaffold. Scale bar is 100 nm.

**Table 3.** Numbers of tubes identified in different *Helicobacter pylori* mutants.
Note that the approximation symbol before the number of cells indicates that in many tomograms we only see a part of the cell(s).

| Mutant | Number of cells | Number of tubes |
|---|---|---|
| *H. pylori ΔfliG fliP** | ~47 | 12 |
| *H. pylori ΔfliM fliP** | ~265 | 88 |
| *H. pylori ΔfliO fliP** | ~267 | 49 |
| *H. pylori ΔfliQ fliP** | ~220 | 55 |
| *H. pylori ΔflgS fliP** | ~84 | 15 |

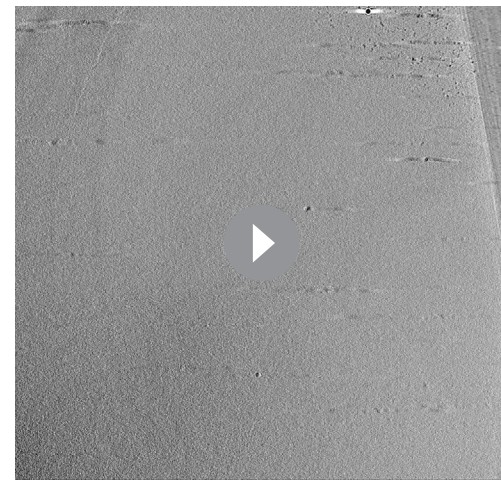

**Video 2.** An electron cryo-tomogram of an *Flavobacterium johnsoniae* cell with outer membrane tubes stemming from the cell. Note the wavy outer membrane of the cell.
https://elifesciences.org/articles/73099/figures#video2

cells. Moreover, while it was previously hypothesized that the formation of membrane tubes in *H. pylori* (when they are in the vicinity of eukaryotic cells) is dependent on the *cag* T4SS (*Chang et al., 2018*), we could not identify any clear correlation between the emanation of membrane tubes and *cag* T4SS particles in our samples where *H. pylori* was not incubated with host cells. We also show that the tubes of *H. pylori* are CORE-independent, indicating that they are different from the CORE-dependent nanotubes described in other species.

A recent study showed that the formation of bacterial tubes significantly increases when cells are stressed or dying (*Pospíšil et al., 2020*). Consistent with this, in our cryo-tomograms we saw many MEs and MVs associated with lysed cells (such as in *H. pylori*, *Helicobacter hepaticus*, and *P. luteoviolacea*). We also saw tubes and vesicles stemming from intact cells. Given the nature of cryo-ET snapshots, we cannot tell whether a cell that appears intact is stressed, nor can we know whether MEs/MVs formed before or after a cell lysed. One observation which might be related to this issue comes from *F. johnsoniae* where tubes with regular diameters were seen stemming mainly from cells with a noticeably wavy OM (45 examples), while pearling tubes and OMVs stemmed primarily from cells with a smooth OM (>100 examples).

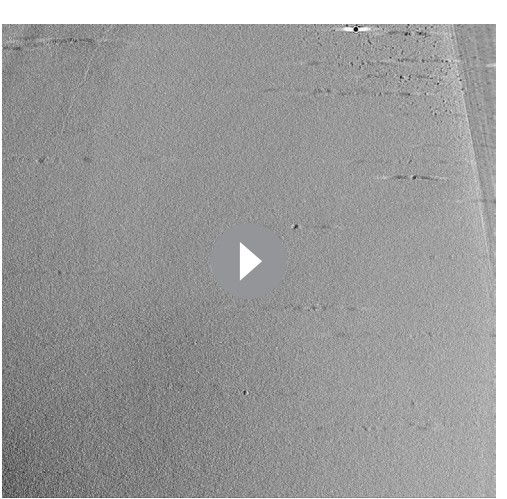

**Video 1.** An electron cryo-tomogram of an *Myxococcus xanthus* cell with multiple outer membrane tubes stemming from the cell.
https://elifesciences.org/articles/73099/figures#video1

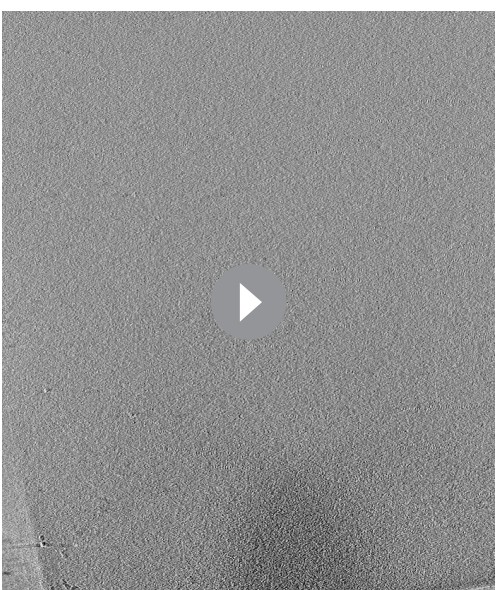

**Video 3.** An electron cryo-tomogram of an *Myxococcus xanthus* cell with a pearling outer membrane tube stemming from the cell.
https://elifesciences.org/articles/73099/figures#video3

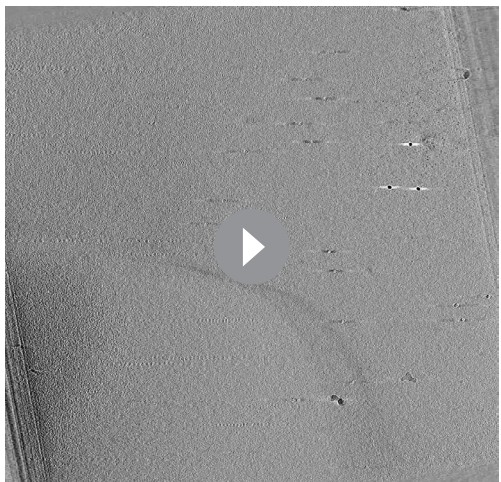

**Video 4.** An electron cryo-tomogram of an *Myxococcus xanthus* cell with multiple branched outer membrane tubes stemming from the cell.
https://elifesciences.org/articles/73099/figures#video4

Compare, for example, the cells in *Figure 1e* and *Figure 7—figure supplement 1* and *Videos 2 and 6* (wavy OM) to those in *Figures 3d and 7a* and f (smooth OM).

In *C. crescentus*, we observed for the first time 'nanopods', a structure previously reported in *D. acidovorans* (*Shetty et al., 2011*). Both of these species are diderms with an S-layer, suggesting that nanopods may be a general form for OMVs in bacteria with this type of cell envelope. Nanopods were proposed to help disperse OMVs in the partially hydrated environment of the soil where *D. acidovorans* lives; it will be interesting to study their function in aquatic *C. crescentus*.

Examining protein complexes associated with OMEs and OMVs, some seemed to reflect a continuation of the same complexes found on the membrane from which the extensions stemmed, such as the T4aP basal body in *M. xanthus* (*Chang et al., 2016*). Others, however, were only observed on MEs and not on cells. This could be because the complexes are related to the formation of the MEs, or it might simply reflect the fact that these extensions are generally thinner and less crowded than the bacterial periplasm, making the complexes easier to see in cryo-tomograms. Interestingly, the crown-like complex we observed in *M. xanthus*, *P. aeruginosa,* and *P. flexibilis* was exclusively associated with the membranes of lysed cells; we never observed it on OMEs and OMVs stemming from intact cells in *M. xanthus*. We observed a morphologically similar crown-like structure with different dimensions in purified naturally shed MEs/MVs of *S. oneidensis*, where we cannot know whether they arose from intact or lysed cells. The crown-like structures are remarkably large and their function remains a mystery. Due to the disruption of membranes in lysed cells, the topology of these complexes is difficult to unravel. However, these structures share a morphological similarity to a membrane-associated dome protein complex recently described on the limiting membrane of the lamellar bodies inside alveolar cells (*Klein et al., 2021*).

Similarly, regarding the different, trapezoidal structure in *S. oneidensis*, the fact that it was seen on both the outside and inside of purified MVs suggests that some of the purified vesicles adopted an inside-out orientation during purification (a documented phenomenon; *Kaplan et al., 2016*). Interestingly, the overall architecture and dimensions of this trapezoidal structure are reminiscent of those of a recently solved structure of the *E. coli* polysaccharide co-polymerase WzzB (*Wiseman et al., 2021*). We hope future investigation by methods like mass spectrometry will characterize these novel ME/MV-associated protein complexes.

In *F. johnsoniae*, we observed secretin-like particles at the tip of ~70 % of tubes stemming from the OM. This strong spatial correlation suggests a role for the secretin-like complex in the formation of MEs in this species. Based on homology, the GspD-like T2SS secretin is a strong candidate for the complex. Interestingly, though, we did not identify any secretin-like (or full T2SS-like) particles in the main cell envelope of *F. johnsoniae* cells. While we could have missed them in the denser periplasm compared to the less-crowded OMEs and OMVs, it is possible that the structures are specifically associated with the formation of OMEs in this species. As these MEs stem only from the OM, there is no IM-embedded energy source for the complex, suggesting that they are not functional secretion systems and raising the

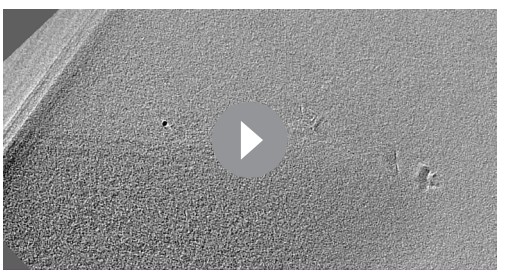

**Video 5.** An electron cryo-tomogram of a *Caulobacter crescentus* cell with a nanopod (black arrow) close to the cell.
https://elifesciences.org/articles/73099/figures#video5

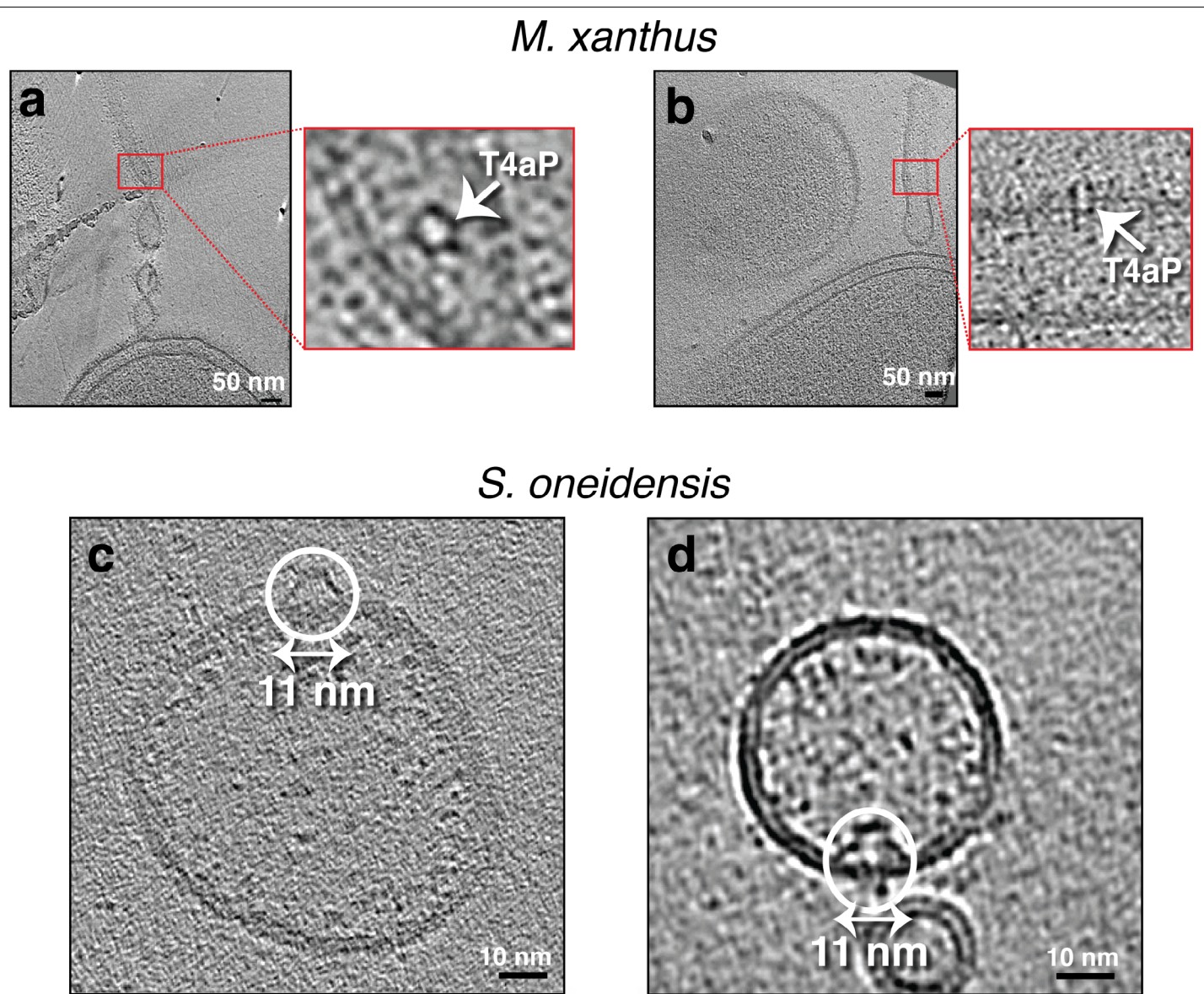

**Figure 5.** Seemingly randomly located protein complexes on outer membrane extensions (OMEs) of *Myxococcus xanthus* and purified membrane vesicles (MVs) of *Shewanella oneidensis*. (**a and b**) Slices through electron cryo-tomograms of *M. xanthus* indicating the presence of pearling tubes with top (**a**) and side (**b**) views of type IVa pilus basal bodies (T4aP). Scale bar is 50 nm. (**c and d**) Slices through electron cryo-tomograms of purified *S. oneidensis* naturally shed MEs and MVs highlighting the presence of trapezoidal structures on the outside (**c**) and inside (**d**) of vesicles. Scale bar is 10 nm.

question of what function they may serve. It is possible that the OMVs and OMEs form to dispense of the secretin.

These complexes also indicate that MEs/MVs may provide an ideal system to investigate membrane-embedded structures in their native environment at higher resolution. For example, it remains unclear how secretins of various secretion systems are situated within the OM. All high-resolution structures were detergent-solubilized, and most in situ structures have low resolution due to cell thickness (*Weaver et al., 2020*). Purifying *F. johnsoniae* OMVs and performing high-resolution subtomogram averaging on the secretin-like complex might shed light on this question.

Early in the history of life, lipid vesicles and elementary protocells likely experienced destabilizing conditions such as repeated cycles of dehydration and rehydration (*Damer and Deamer, 2015*). The binding of prebiotic amino acids to lipid vesicles can help stabilize them in such conditions (*Cornell*

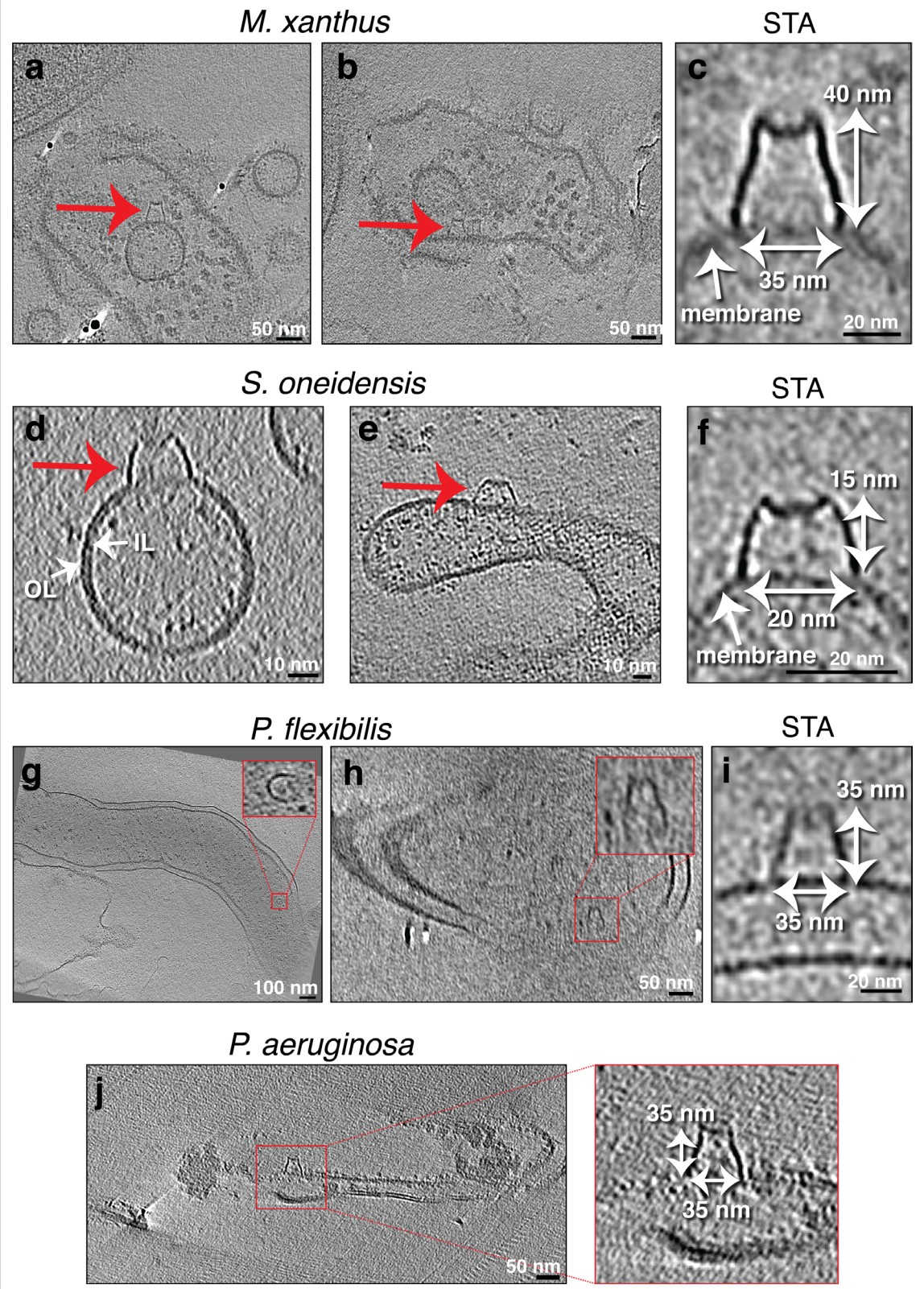

**Figure 6.** Seemingly randomly located protein complexes associated with lysed cells. Slices through electron cryo-tomograms of lysed cells (**a, b, g, h, and j**) or purified membrane extensions (MEs) and membrane vesicles (MVs) (**d and e**) showing the presence of MVs and lysed membranes with a crown-like complex (red arrows and red boxed enlargements). Scale bars: 50 nm (**a, b, h, and j**), 100 nm (**g**), 10 nm (**d and e**). (**c, f, and i**) Central slices through subtomogram averages (with twofold symmetry along the Y-axis applied) of nine particles (**c**), four particles, (**f**), or three particles (**i**) of the crown-like

*Figure 6 continued on next page*

*Figure 6 continued*

complex in the indicated species. Scale bar is 20 nm. OL = outer leaflet, IL = inner leaflet.

The online version of this article includes the following figure supplement(s) for figure 6:

**Figure supplement 1.** Slices through electron cryo-tomograms of lysed *Pseudomonas aeruginosa* cells indicating the presence of crown-like structures in side views (**a** and **b**) and top view (**c**, dashed yellow ellipses).

*et al., 2019*) and it is conceivable that with billions of years of evolution, variations of these stabilized lipid structures acquired roles that conferred fitness advantages on bacterial species in various environments. Today, the ability of bacteria to extend their membranes to form tubes or vesicles is a widespread phenomenon with many important biological functions. We hope that the structural classification we present here will serve as a helpful reference for future studies in this growing field.

## Materials and methods
### Strains and growth conditions
*Hylemonella gracilis* cells were grown as described in *Kaplan et al., 2020*. *P. luteoviolacea* were grown as described in *Shikuma et al., 2014*. *Magnetospirillum magneticum* were grown as described in *Cornejo et al., 2016*. *P. flexibilis* 706570 were grown in lactose growth medium. *S. oneidensis* MR-1 cells were grown, as detailed in *Phillips et al., 2020*, in Luria Bertani (LB) media under aerobic conditions at 30 °C with shaking at 200 rpm until they reached $OD_{600}$ of ~3. *M. xanthus* PilY1.3-sfGFP, *M. xanthus* ΔtsaP, and *M. xanthus* SA6892 strains were grown as described in *Chang et al., 2016*. *B. burgdorferi* B31 ATCC 35210 and *H. hepaticus* ATCC 51,449 cells were grown in standard media (see *Briegel et al., 2009* and references therein).

*C. crescentus* was cultured in M2G and M2 media (prepared as described in *Schrader and Shapiro, 2015*); 5 mL of M2G were inoculated with a frozen stock of *C. crescentus* NA 1000 (wild-type and D*pleD* mutant cells; see *Kaplan et al., 2021c*) and grown overnight at 28°C; and 5 mL of the overnight culture was diluted in 15 mL M2G and grown at 28°C with a shaking speed of 200 rpm for ~2 hr until mid-log phase ($OD_{600}$ 0.4–0.5). The sample was then centrifuged at 5200 × *g* for 6 min at 4°C (same temperature for all subsequent centrifugation steps) and the pellet was resuspended in 1 mL M2 solution. The resuspended cells were transferred into a 2 mL microcentrifuge tube and centrifuged at 5200 × *g* for 5 min. All but ~250 μL of supernatant was removed, 650 μL M2 was added and the pellet was resuspended, and 900 μL cold Percoll (Sigma Aldrich) was added and the sample was centrifuged at 15,000 × *g* for 20 min. Samples were taken from the bottom of the tube to select swarmer cells.

Cells of *F. johnsoniae* strain CJ2618 (a wild-type strain overexpressing FtsZ, ATCC 17061) were taken from a glycerol stock, streaked onto a CYE plate with 10 μg/mL tetracycline and grown at 25 °C. Subsequently, 5 mL of motility medium (MM) was inoculated with colonies from the plate and the culture was incubated at 25 °C with 80 rpm shaking overnight. Then another 5 mL MM was inoculated with 80 μL of starter culture and placed at 25 °C with no shaking until the next day when the cells were harvested and prepared for plunge-freezing.

*H. pylori* mutants (Δ*fliM fliP\**, Δ*fliO fliP\**, Δ*flgS fliP\**, Δ*fliG fliP\**, Δ*fliQ fliP\**) were grown from glycerol stocks on sheep blood agar at 37 °C with 5 % $CO_2$ for 48 hr and then either plunge-frozen directly or the cells were spread on another plate and left to grow for 24 hr before plunge-freezing. No difference could be discerned between the two samples by cryo-ET.

*F. anhuiense* (strain 98, see *Carrión et al., 2019*) and *C. pinensis* (strain 94, see *Carrión et al., 2019*) cells were grown overnight in 1/10 TSB at 25 °C and 300 rpm shaking in 50 mL cultures. For sample preparation, cells were first concentrated by centrifugation; 3 μL aliquots of the cell suspension were applied to glow-discharged R2/2, 200 mesh copper Quantifoil grids (Quantifoil Micro Tools), the sample was pre-blotted for 30 s, and then blotted for 2.5 s (*F. anhuiense*) and 1 s (*C. pinensis*). Grids were pre-blotted and blotted at 20 °C and at 95 % humidity. Subsequently, the grids were plunge-frozen in liquid ethane using an automated Leica EM GP system (Leica Microsystems) and stored in liquid nitrogen.

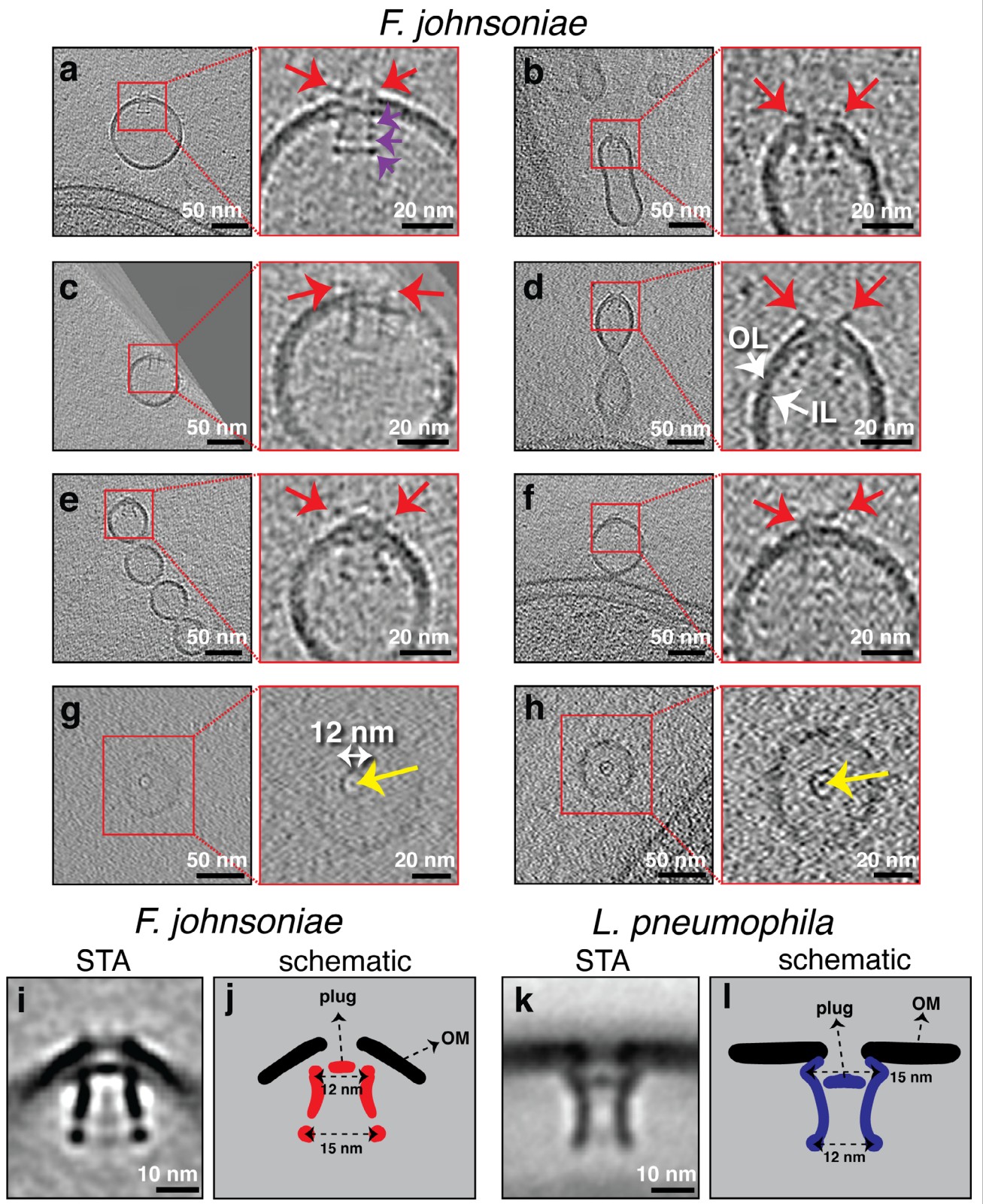

**Figure 7.** Secretin-like complexes located at the tip of outer membrane extensions (OMEs) and outer membrane vesicles (OMVs) in *Flavobacterium johnsoniae*. Slices through electron cryo-tomograms of *F. johnsoniae* illustrating the presence of secretin-like complexes (side views in **a–f**), top views in (**g and h**) with yellow arrows pointing to the plug in OMEs and OMVs of *F. johnsoniae*. Red arrows point to the extracellular part of the complex. Purple arrows in the enlargement in (**a**) point to the three periplasmic densities. Scale bars are 50 nm in main panels and 20 nm in enlargements. (**i**) A

*Figure 7 continued on next page*

Figure 7 continued

central slice through the subtomogram average of 88 particles of the secretin-like complex (with twofold symmetry along the Y-axis applied). Scale bar is 10 nm. (**j**) A schematic representation of the STA shown in (**i**). (**k**) A central slice through the subtomogram average of the secretin of the type II secretion systems (T2SS) of *Legionella pneumophila* (EMD 20713, see *Ghosal et al., 2019*). Scale bar is 10 nm. (**l**) A schematic representation of the STA shown in (**k**).

The online version of this article includes the following figure supplement(s) for figure 7:

**Figure supplement 1.** Slices through electron cryo-tomograms of *Flavobacterium johnsoniae* (with wavy outer membrane [OM]) illustrating tubes stemming from cells with secretin-like complexes at their tips, as highlighted in the enlargements on the right (white circles).

**Figure supplement 2.** FSC curves of the subtomogram average of the secretin-like complex.

## Purification of *S. oneidensis* OMVs

*S. oneidensis* OMVs were purified as described in *Phillips et al., 2020*. First, *S. oneidensis* were grown in LB media until they reached $OD_{600}$ of 3. Subsequently, the cells were centrifuged at $5000 \times g$ for 20 min at 4 °C; the pellet contained whole cells while the supernatant contained the OMVs. To remove any cells present in the supernatant, it was filtered through a 0.45 µm filter. Subsequently, the supernatant was centrifuged at $38,400 \times g$ for 1 hr at 4 °C; the OMVs were in the resultant pellet. The pellet was resuspended in 20 mL of 50 mM HEPES pH 6.8 buffer, filtered through a 0.22 µm filter, spun again as described above, and ultimately resuspended in 50 mM HEPES pH 6.8.

## Cryo-ET sample preparation and imaging

For cellular samples, 10 nm gold beads were first coated with BSA (bovine serum albumin) and then mixed with the cells. Subsequently, 4 µL of this mixture was applied to a glow-discharged, thick carbon-coated, R2/2, 200 mesh copper Quantifoil grid (Quantifoil Micro Tools) in an FEI Vitrobot chamber with 100 % humidity. Excess fluid was blotted away with filter paper and the grid was plunge-frozen in a mixture of ethane/propane. For the purified OMVs of *S. oneidensis*, the sample was first diluted to a 0.4 mg/mL concentration before it was applied to the grid (*Phillips et al., 2020*). Cryo-ET imaging of the samples was done either on an FEI Polara 300 keV field emission gun transmission electron microscope equipped with a Gatan imaging filter and a K2 Summit direct electron detector in counting mode, or a Thermo Fisher Titan Krios 300 keV field emission gun transmission electron microscope equipped with a Gatan imaging filter and a K2 Summit counting electron detector. For data collection, either the UCSF Tomography (*Zheng et al., 2007*) or SerialEM (*Mastronarde, 2005*) software was used. For OMVs, tilt series spanned –60° to 60° with an increment of 3°, an underfocus of 1–5 µm, and a cumulative electron dose of 121 e/Å². For *F. johnsoniae*, tilt series spanned –55° to 55° with 1° increment, an underfocus of 4 µm, a cumulative electron dose of 100 e/Å², and a 3.9 Å pixel size. For *M. xanthus,* tilt-series spanned –60° to 60° with an increment of 1°, an underfocus of 6 µm, and a cumulative electron dose of 180 e/Å². For *B. burgdorferi*, tilt series spanned –60° to 60° with 1° increment, an underfocus of 10 µm, and a cumulative electron dose of 160 e/Å². For *H. hepaticus*, tilt series spanned –60° to 60° with increments of 1°, an underfocus of 12 µm, and a cumulative electron dose of 165 e/Å².

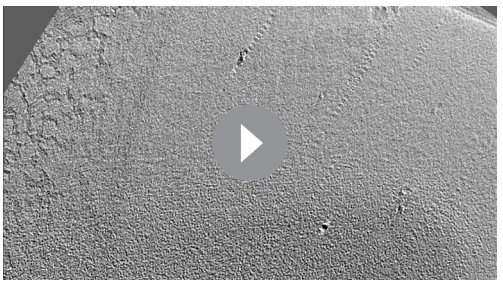

**Video 6.** An electron cryo-tomogram of an *Flavobacterium johnsoniae* cell highlighting the presence of secretin-like particles at the tips of outer membrane tubes.

https://elifesciences.org/articles/73099/figures#video6

*F. anhuiense* and *C. pinensis* images were recorded with a Gatan K3 Summit direct electron detector equipped with a Gatan GIF Quantum energy filter with a slit width of 20 eV. Images were taken at magnification corresponding to a pixel size of 3.28 Å (*C.pinensis*) and 4.4 Å (*F. anhuiense*). Tilt series were collected using Seri-alEM with a bidirectional dose-symmetric tilt scheme (–60° to 60°, starting from 0°) with a 2° increment. The defocus was set to – 8 to 10 µm and the cumulative exposure per tilt series was 100 e⁻/A² was. Images were reconstructed with the IMOD software package.

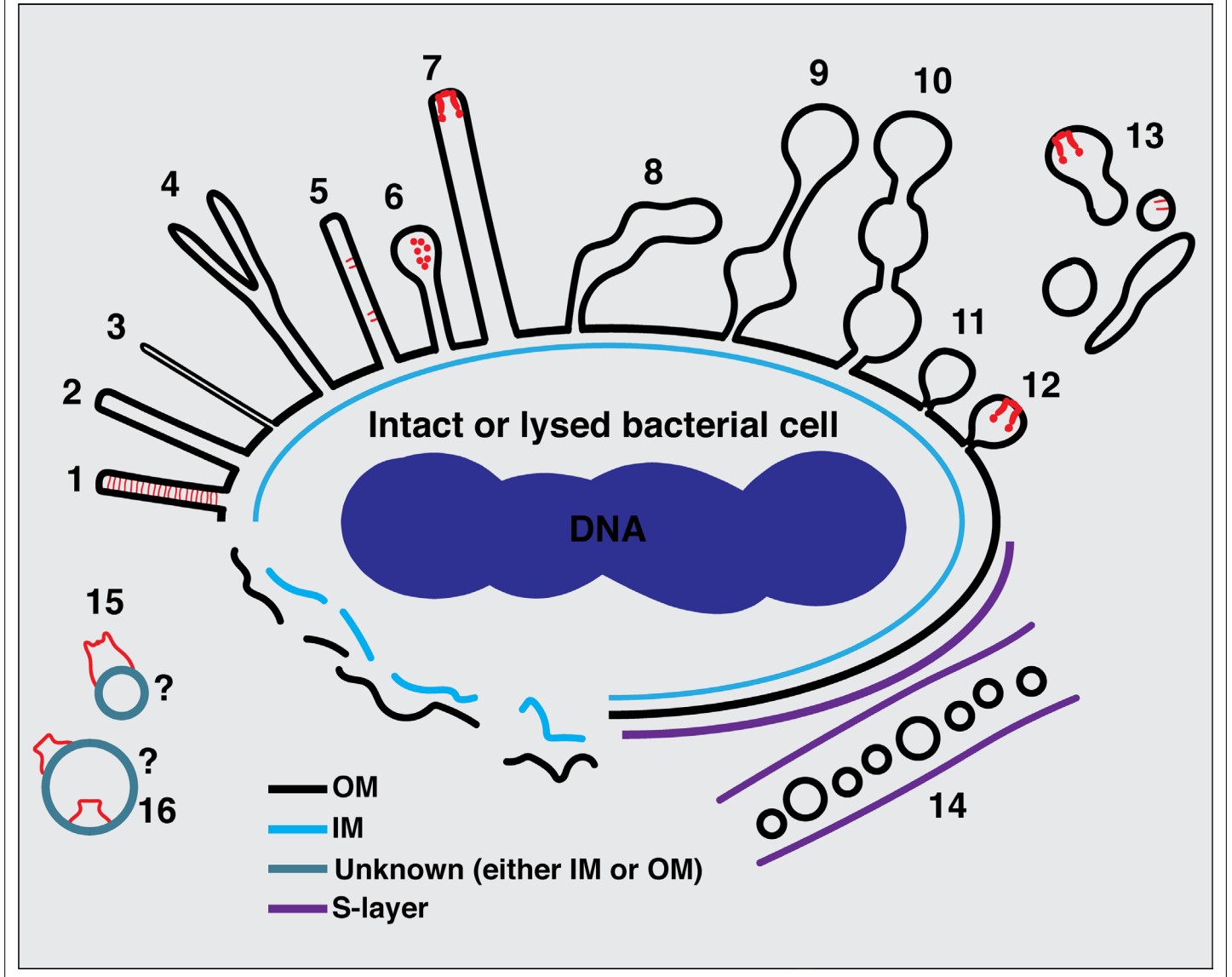

**Figure 8.** Summary of types of membrane extensions (MEs) and membrane vesicles (MVs) identified in this study. (1) Tubes with a uniform diameter and with an internal scaffold; (2 and 3) tubes with a uniform diameter but without an internal scaffold; (4) bifurcating tubes; (5) tubes with seemingly randomly located protein complexes (type IVa pilus [T4aP]); (6) teardrop-like extensions; (7) tubes with a secretin-like complex at their tip; (8) tubes with irregular diameter; (9) pearling tubes; (10) interconnected chains of vesicles with 14 nm connectors; (11) budding vesicles; (12) budding vesicles with a secretin-like complex at their tip; (13) various disconnected membrane structures in the vicinity of bacterial cells; (14) nanopods in species with an inner membrane (IM), outer membrane (OM), and S-layer; (15) membrane structures with a crown-like complex from lysed cells; (16) purified outer MVs (OMVs) with trapezoidal complexes. The question marks in (15) and (16) indicate the difficulty of determining whether a membrane structure from lysed cells or purified vesicles originated from the IM or the OM or is in its original topology.

## Image processing and subtomogram averaging

Reconstruction of tomograms of cellular samples was done using the automatic RAPTOR pipeline implemented in the Jensen lab at Caltech (*Ding et al., 2015*). Tomograms of purified *S. oneidensis* OMVs were reconstructed using a combination of ctffind4 (*Rohou and Grigorieff, 2015*) and the IMOD software package (*Kremer et al., 1996*). Subtomogram averaging was done using the PEET program (*Nicastro, 2006*), with twofold symmetry applied along the particle Y-axis.

## Acknowledgements

This project was funded by the NIH (grant R35 GM122588 to GJJ, and P20 GM130456 to CLS) and a Baxter postdoctoral fellowship from Caltech to MK. Cryo-ET work was done in the Beckman Institute Resource Center for Transmission Electron Microscopy at the California Institute of Technology. We are grateful to Prof. Martin Pilhofer for collecting the *P. luteoviolacea* data and for critically reading the manuscript. We thank Prof. Elitza I Tocheva for collecting the *D. acidovorans* data. We thank Prof. Mohamed El-Naggar for insights into preparing *S. oneidensis* samples and Dr. Yuxi Liu for discussions. Briegel lab data was collected at the Netherlands Center for Electron Nanoscopy with support from Dr Wen Yang. This data was collected with support from the National Roadmap for Large-Scale Research Infrastructure 2017–2018 with project number 184.034.014, which is financed in part by the Dutch Research Council (NWO). This work was also supported by the NWO OCENW. GROOT.2019.063 grant.

## Additional information

### Funding

| Funder | Grant reference number | Author |
| --- | --- | --- |
| National Institutes of Health | R35 GM122588 | Grant J Jensen |
| California Institute of Technology | Baxter postdoctoral fellowship | Mohammed Kaplan |
| Nederlandse Organisatie voor Wetenschappelijk Onderzoek | NWO OCENW. GROOT.2019.063 | Ariane Briegel |
| National Institutes of Health | P20 GM130456 | Carrie L Shaffer |
| Nederlandse Organisatie voor Wetenschappelijk Onderzoek | 184.034.014 | Ariane Briegel |

The funders had no role in study design, data collection and interpretation, or the decision to submit the work for publication.

### Author contributions

Mohammed Kaplan, Conceptualization, Data curation, Formal analysis, Funding acquisition, Writing - original draft; Georges Chreifi, Lauren Ann Metskas, Cecily R Wood, Poorna Subramanian, Lori A Zacharoff, Yuhang Wang, Yi-Wei Chang, Morgan Beeby, Megan J Dobro, Yongtao Zhu, Mark J McBride, Data curation, Writing – review and editing; Janine Liedtke, Data curation, Formal analysis, Writing – review and editing; Catherine M Oikonomou, William J Nicolas, Formal analysis, Writing – review and editing; Ariane Briegel, Formal analysis, Funding acquisition, Writing – review and editing; Carrie L Shaffer, Data curation, Formal analysis, Funding acquisition, Writing – review and editing; Grant J Jensen, Conceptualization, Formal analysis, Funding acquisition, Investigation, Supervision, Writing – review and editing

### Author ORCIDs

Mohammed Kaplan http://orcid.org/0000-0002-0759-0459
Georges Chreifi http://orcid.org/0000-0003-4194-1694
Lauren Ann Metskas http://orcid.org/0000-0002-8073-6960
Janine Liedtke http://orcid.org/0000-0003-2680-4130
Catherine M Oikonomou http://orcid.org/0000-0003-2312-4746
William J Nicolas http://orcid.org/0000-0001-5970-8626
Yuhang Wang http://orcid.org/0000-0003-3715-8349
Yi-Wei Chang http://orcid.org/0000-0003-2391-473X
Morgan Beeby http://orcid.org/0000-0001-6413-9835
Megan J Dobro http://orcid.org/0000-0002-6464-3932
Yongtao Zhu http://orcid.org/0000-0002-3069-6518
Mark J McBride http://orcid.org/0000-0002-3798-6761

Ariane Briegel http://orcid.org/0000-0003-3733-3725
Carrie L Shaffer http://orcid.org/0000-0002-7457-7422
Grant J Jensen http://orcid.org/0000-0003-1556-4864

### Decision letter and Author response

Decision letter https://doi.org/10.7554/eLife.73099.sa1
Author response https://doi.org/10.7554/eLife.73099.sa2

## Additional files

### Supplementary files
• Transparent reporting form

### Data availability
All data generated or analysed during this study are included in the manuscript and supporting files and movies.

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
