## [Decision Letter]

[Editors' note: this paper was reviewed by Review Commons.]

**Acceptance summary:**

In this study, the authors survey a plethora of bacterial outer-membrane projections captured over the years by in situ cryo-tomography under near-native conditions. They classify visualized structures, highlighting both similarities and differences among them and further describe molecular complexes that are associated with these projections. The manuscript highlights the abundance of such understudied structures in nature, indicating the need to deepen the exploration into their biological functions and mechanisms of action. This work will be of interest to microbiologists in general.

---

## [Author Response]

We thank the editors of Review Commons and all the reviewers for their insightful comments which helped us to improve our manuscript. We have now modified our manuscript based on the Reviewers’ comments and would like to ask you to consider our revised manuscript for publication.

Reviewer #1:This manuscript by the Jensen lab surveys a plethora of bacterial outer-membrane projections captured over the years by in situ cryo-tomography under near-native conditions. The authors classify the different visualized structures, highlighting both similarities and differences among them. They further describe molecular complexes that are associated with these projections. The manuscript highlights the abundance of such understudied structures in nature, indicating the need to deepen our exploration into their biological functions and mechanisms of action.

We thank the reviewer for her/his insightful comments that allowed us to improve our manuscript.

1. The authors should state in the Abstract and Introduction that only diderm bacteria and outermembrane extensions are included in the study.

Done. We have modified the title, the abstract and the introduction to explicitly highlight this point.

2. In the Introduction or Discussion the authors should mention the limits of the in situ cryo-tomography, such as the difficulty to observe regions in between neigbouring bacterial cells, and into the thick bacterial cell body.

Done. We have added the following to our revised manuscript:

“Currently, only electron cryo-tomography (cryo-ET) allows visualization of structures in a near-native state inside intact (frozen-hydrated) cells with macromolecular (~5 nm) resolution. However, this capability is limited to thin samples (few hundred nanometers thick, like individual bacterial cells of many species) while thicker samples like the central part of eukaryotic cells, thick bacterial cells, or clusters of bacterial cells are not amenable for direct cryo-ET imaging. Such thick samples can be rendered suitable for cryo-ET experiments by thinning them first using different methods including focused ion beam milling and cryosectioning [30]. Cryo-ET has already been invaluable in revealing the structures of several membrane extensions, including *Shewanella oneidensis* nanowires [6], *Helicobacter pylori* tubes [15], *Delftia acidovorans* nanopods [25], *Vibrio vulnificus* OMV chains [16], and more recently cell-cell bridges in the archaeon *Haloferax volcanii* [31].”

3. Please provide a legend to Table S1 explaining the numbers (organelles?), how many cells were viewed? I think that at least part of it should be included in the main text. Also, there are examples of vesicles emanating from H. pylori. This information is missing from Table S1.

Done. We added a column to the table indicating the number of cells available for each species. We also added the information about the vesicles in *H. pylori* to the table. This table is now incorporated into the main text of the manuscript as Table 1.

4. Please provide an ordered list including all the strains (and IDs of the specific isolates) used in this study and their genotypes.

Done. We added Table S1 to the revised manuscript that contains this information. This table also includes relevant references to all the published papers where these strains were previously used.

5. The authors describe in detail the H. pylori tubes that seem to be flagellum-core independent. However, the authors found previously (ref 15) that during infection, these structures are dependent on CagA T4SS, and they visualized T4SS sub-complexes in proximity to the point of tube emanation. This should be described and discussed in the text. Also, please indicate if the "host-independent" tubes are similarly dependent on T4SS.

Done. We added the following to the revised manuscript:

“The scaffolded uniform tubes of *H. pylori* that we observed were formed in samples not incubated with eukaryotic cells, indicating that they can also form in their absence. However, the tubes we found had closed ends and no clear lateral ports, while some of the previously-reported tubes (formed in the presence of eukaryotic host cells) had open ends and prominent ports [15]. It is possible that such features are formed only when *H. pylori* are in the vicinity of host cells. Moreover, while it was previously hypothesized that the formation of membrane tubes in *H. pylori* (when they are in the vicinity of eukaryotic cells) is dependent on the *cag* T4SS [15], we could not identify any clear correlation between the emanation of membrane tubes and *cag* T4SS particles in our samples where *H. pylori* was not incubated with host cells. We also show that the tubes of *H. pylori* are CORE-independent, indicating that they are different from the CORE-dependent nanotubes described in other species.”

6. Is there any difference in the frequency or length of the tubes in the mutants presented in Figure 4? The flgS mutant in the image exhibits a very short filament; is that typical?

We did not see any significant statistical difference in the number or lengths of the tubes in these different mutants. We added Table S2 to the revised manuscript which details the number of cells we visualized for each mutant and the number of the tubes seen there. In all these mutants the lengths of the tubes ranged between few tens to hundreds of nanometers. In addition, we added Figure S2 to show more examples of these tubes in each of these mutants.

Minor points:– Please check full bacterial names that are sometimes missing (e.g., lines 110-112).

Done.

– There is no reference to panel 2G. Please check the references to all panels.

Done. Please see lines 154 and 183 in the main text.

– Lines 181-184: There is no figure related to the formation of teardrop-like extensions from C. pinensis. Please review the text accordingly.

Done. Corrected.

– Line 235, not clear to what "as these" refers to.

Done. We modified the text as the following:

“As these MEs/MVs from *S. oneidensis* were purified”

– Line 241, not clear what "a secretin-like complex" is, and no reference is provided.

Done. We modified the text as the following:

“In the third category, we observed a secretin-like complex in many tubes and vesicles of *F. johnsoniae*. Secretins are proteins that form a pore in the outer membrane and are associated with many secretion systems like type IV pili and type II secretion systems (T2SS) [39–41]”

Reviewer #1 (Significance (Required)):As described in this manuscript, even in model bacteria these structures are generated (e.g., Caulobacter forms the hardly studied nanopod extensions). The manuscript also provides visual categories of these structures, defining "extension types" that are likely to be used by the scientific community for years to come, similar to the initial pili classification during the 1960s-70s. It is a "descriptive study," in the positive sense of the term, as it significantly contributes to the field of bacteriology.

We thank the reviewer for her/his kind words and enthusiasm about our work. It is an honor to have our work compared to the seminal pili classification work done in the 1960s-70s by pioneers in the field of bacteriology.

Reviewer #2:The manuscript "In situ imaging of bacterial membrane projections and associated protein complexes using electron cryo-tomography" by Kaplan et al., identifies and catalogues membrane extensions (MEs) and membrane vesicles (MVs) from 13 different species using cryo-electron tomography. Furthermore, they identify and discuss several protein complexes observed in these membrane projections.The manuscript is beautifully written, interesting, and genuinely got this reviewer excited about the biology. I applaud the authors on their manuscript and have only minor comments and a few thoughts that the authors may wish to think on and discuss.

We thank the reviewer for her/his kind words and insightful comments that allowed us to improve our manuscript.

– Some schematics throughout the introduction would be useful to readers new to the field/ outside the field who are not used to these different membrane structure features.

We thank the Reviewer for this suggestion. First, we made an extra figure with schematics showing the cell body and membrane tubes but that was rather redundant with Figure 8. For this reason, we added explicit labels to figure 1 highlighting the cell body and the tubes in these examples to help the reader following that figure and the subsequent ones. However, if the Reviewer has an explicit suggestion/view about the schematics then we would be very happy to do that.

– The size of scale bars should be indicated on the figure panels themselves rather than in the figure legend to assist the reader.

Done.

– In reference to lines 193-196 – what was the extracellular environment like in these micrographs? Were other cells present? Could it be the extracellular environment/surrounding cells that stimulate pearling? Have the authors considered this? Please discuss if relevant/insightful.

This is a good point. The cells were usually plunge-frozen in their standard growth media (except in *H. pylori* where the cells were resuspended in PBS and subsequently plunge-frozen). Yes, there are other cells present in the sample, however, usually, only one cell is present in the field of view of the tomogram as areas with multiple cells have thick ice and therefore not amenable for cryo-ET imaging. We added the following to the revised manuscript:

“As usually only one (or part of a) cell is present in the cryo-tomogram, we can’t exclude that differences in the extracellular environments, like the presence of a cluster of cells in the vicinity of the individual cells with pearling tubes, might play a role in this observation”.

– "Randomly-located complexes" in this reviewers opinion should actually be described "seemingly randomly-located complexes" given there may be an organization present that is beyond the resolution limit of this study.

The is a good point. Indeed, we can’t exclude that these complexes have a preferred localization in specific lipid patches that we can’t detect in our cryo-tomograms. We added the following statement to the revised manuscript:

“These complexes, which were also found in the OM of intact cells, did not exhibit a preferred localization or regular arrangement within the tube at least within the fields of view provided by our cryotomograms (Figure 5a & b).”.

– In reference to lines 287-292 – is it possible this has to do with lipid composition? Have the authors considered this? Please discuss if relevant/insightful.

Done. We added the following to the revised manuscript:

“In addition, differences in the lipid compositions among the various species investigated here might also play a role in the formation of these different forms of projections”.

Reviewer #2 (Significance (Required)):These results advance the field by shedding new light on bacterial membrane extension morphologies. The authors use a cryo-ET to catalogues membrane extensions and membrane vesicles which has not been done before.This paper is likely to be of interest to structural biologists, biophysicist, membrane protein biologists, virologists and microbiologists.This reviewer is a single-particle cryo-EM structural biologist with interest in membrane proteins.

We thank the reviewer for her/his enthusiasm about our work described here.